# Skill-based Model-based Reinforcement Learning

**Lucy Xiaoyang Shi**[1]    **Joseph J. Lim**[2,3*]    **Youngwoon Lee**[1]

[1]University of Southern California    [2]KAIST    [3]NAVER AI Lab

**Abstract:** Model-based reinforcement learning (RL) is a sample-efficient way of learning complex behaviors by leveraging a learned single-step dynamics model to plan actions in imagination. However, planning every action for long-horizon tasks is not practical, akin to a human planning out every muscle movement. Instead, humans efficiently plan with high-level skills to solve complex tasks. From this intuition, we propose a Skill-based Model-based RL framework (SkiMo) that enables planning in the skill space using a *skill dynamics model*, which directly predicts the skill outcomes, rather than predicting all small details in the intermediate states, step by step. For accurate and efficient long-term planning, we *jointly* learn the skill dynamics model and a skill repertoire from prior experience. We then harness the learned skill dynamics model to accurately simulate and plan over long horizons in the skill space, which enables efficient downstream learning of long-horizon, sparse reward tasks. Experimental results in navigation and manipulation domains show that SkiMo extends the temporal horizon of model-based approaches and improves the sample efficiency for both model-based RL and skill-based RL. Code and videos are available at `https://clvrai.com/skimo`.

**Keywords:** Model-Based Reinforcement Learning, Skill Dynamics Model

## 1 Introduction

A key trait of human intelligence is the ability to plan abstractly for solving complex tasks [1]. For instance, we perform cooking by imagining outcomes of high-level skills like washing and cutting vegetables, instead of planning every muscle movement involved [2]. This ability to plan with temporally-extended skills helps to scale our internal model to long-horizon tasks by reducing the search space of behaviors. To apply this insight to artificial intelligence agents, we propose a novel skill-based and model-based reinforcement learning (RL) method, which learns a model and a policy in a high-level skill space, enabling accurate long-term prediction and efficient long-term planning.

Typically, model-based RL involves learning a flat single-step dynamics model, which predicts the next state from the current state and action. This model can then be used to simulate "imaginary" trajectories, which significantly improves sample efficiency over their model-free alternatives [3, 4]. However, such model-based RL methods have shown only limited success in long-horizon tasks due to inaccurate long-term prediction [5] and computationally expensive search [6, 7, 8].

Skill-based RL enables agents to solve long-horizon tasks by acting with multi-action subroutines (skills) [9, 10, 11, 12, 13, 14] instead of primitive actions. This temporal abstraction of actions enables systematic long-range exploration and allows RL agents to plan farther into the future, while requiring a shorter horizon for policy optimization, which makes long-horizon downstream tasks more tractable. Yet, on complex long-horizon tasks, skill-based RL still requires a few million to billion environment interactions to learn [13], which is impractical for real-world applications.

To combine the best of both model-based RL and skill-based RL, we propose **Ski**ll-based **Mo**del-based RL (**SkiMo**), which enables effective planning in the skill space using a *skill dynamics model*. Given a state and a skill to execute, the skill dynamics model directly predicts the resultant state after skill execution, without needing to model every intermediate step and low-level action (Figure 1), whereas the flat dynamics model predicts the immediate next state after one action execution. Thus,

---

[*]AI Advisor at NAVER AI Lab

6th Conference on Robot Learning (CoRL 2022), Auckland, New Zealand.

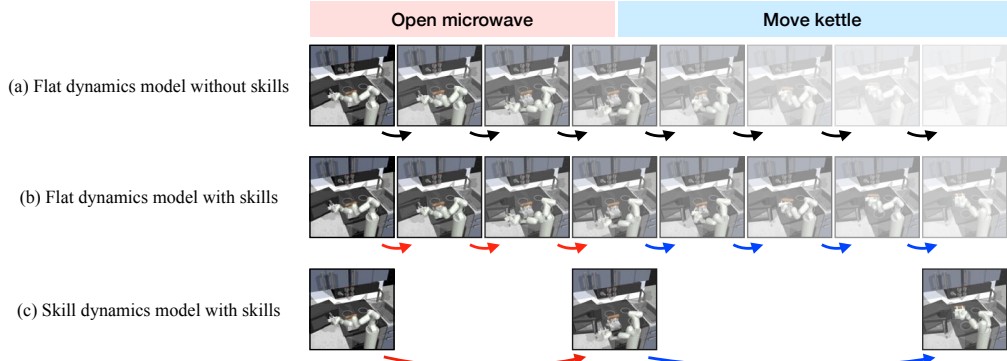

Figure 1: Intelligent agents can use their internal models to imagine potential futures for planning. Instead of planning out every primitive action (black arrows in **a**), they aggregate action sequences into skills (red and blue arrows in **b**). Further, they can leap directly to the predicted outcomes of executing skills in sequence (red and blue arrows in **c**), which leads to better long-term prediction and planning compared to predicting step-by-step (blurriness of images represents the level of error accumulation in prediction).

planning with skill dynamics requires fewer predictions than flat dynamics, resulting in more reliable long-term future predictions and plans.

Concretely, we first jointly learn the skill dynamics model and a skill repertoire from large offline datasets collected across diverse tasks [15, 12, 16]. This joint training shapes the skill embedding space for easy skill dynamics prediction and skill execution. Then, to solve a complex downstream task, we train a high-level task policy that acts in the learned skill space. For more efficient policy learning and better planning, we leverage the skill dynamics model to simulate skill trajectories.

The main contribution of this work is to propose *Skill-based Model-based RL (SkiMo)*, a novel sample-efficient model-based hierarchical RL algorithm that leverages task-agnostic data to extract not only a reusable skill set but also a skill dynamics model. The skill dynamics model enables efficient and accurate long-term planning for sample-efficient RL. Our experiments show that our method outperforms the state-of-the-art skill-based and model-based RL algorithms on long-horizon navigation and robotic manipulation tasks with sparse rewards.

## 2 Related Work

Model-based RL leverages a (learned) dynamics model of the environment to plan a sequence of actions that leads to the desired behavior. The dynamics model predicts the future state of the environment after taking a specific action, which enables simulating candidate behaviors in imagination instead of in the physical environment. Then, these imaginary rollouts can be used for planning [8, 4] through, e.g., CEM [17] and MPPI [18], as well as for policy optimization [19, 3, 20, 4] to improve sample efficiency. Yet, due to the accumulation of prediction error at each step and the increasing search space, finding an optimal, long-horizon plan is inaccurate and computationally expensive [6, 7, 8].

To facilitate learning of long-horizon behaviors, skill-based RL lets the agent act over temporally-extended skills (i.e. options [21] or motion primitives [22]), which can be represented as sub-policies or a coordinated sequence of low-level actions. Temporal abstraction effectively reduces the task horizon for the agent and enables directed exploration [23]. The reusable skills can be manually defined [22, 24, 10, 11, 14], extracted from large offline datasets [15, 25, 26, 27, 28], discovered online in an unsupervised manner [29, 30], or acquired in the form of goal-reaching policies [31, 32, 33, 34, 20]. However, skill-based RL is still impractical for real-world applications, requiring a few million to billion environment interactions [13]. In this paper, we use model-based RL to guide the planning of skills to improve the sample efficiency of skill-based approaches.

There have been attempts to plan over skills in model-based RL [29, 35, 5, 36, 37]. However, most of these approaches [29, 5, 36] still utilize the conventional flat (single-step) dynamics model, which

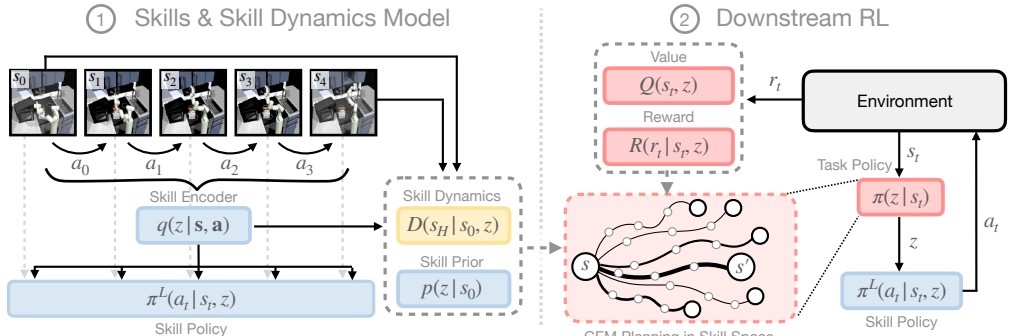

Figure 2: Our approach, *SkiMo*, combines model-based RL and skill-based RL for sample efficient learning of long-horizon tasks. *SkiMo* consists of two phases: (1) learn a skill dynamics model and a skill repertoire from offline task-agnostic data, and (2) learn a high-level policy for the downstream task by leveraging the learned model and skills. We omit the encoded latent state $\mathbf{h}$ in the figure and directly write the observation $\mathbf{s}$ for clarity, but most modules take the latent state $\mathbf{h}$ as input.

struggles at handling long-horizon planning due to error accumulation. Wu et al. [35] proposes to learn a temporally-extended dynamics model; however, it conditions on low-level actions rather than skills and is only used for low-level planning. A concurrent work, Shah et al. [37], is most similar to our work in that it learns a skill dynamics model, but with a limited set of discrete, manually-defined skills. To fully unleash the potential of temporally abstracted skills, we extract the skill space from data and devise a skill-level dynamics model to provide accurate long-term prediction, which is essential for solving long-horizon tasks. To the best of our knowledge, *SkiMo* is the first work that jointly learns skills and a skill dynamics model from data for model-based RL.

## 3 Method

To enable accurate long-term prediction and efficient long-horizon planning for RL, we introduce *SkiMo*, a novel skill-based and model-based RL algorithm that shares synergistic benefits from both frameworks. A key change to prior model-based approaches is the use of a *skill dynamics model* that directly predicts the outcome of a chosen skill, which enables efficient and accurate long-term planning. As illustrated in Figure 2, our approach consists of two phases: (1) learning the skill dynamics model and skills from an offline dataset (Section 3.3) and (2) downstream task learning with the skill dynamics model (Section 3.4).

### 3.1 Preliminaries

**RL**   We formulate a problem as a Markov decision process [38], which is defined by a tuple $(\mathcal{S}, \mathcal{A}, R, P, \rho_0, \gamma)$ of the state space $\mathcal{S}$, action space $\mathcal{A}$, reward $R(\mathbf{s}, \mathbf{a})$, transition probability $P(\mathbf{s}'|\mathbf{s}, \mathbf{a})$, initial state distribution $\rho_0$, and discounting factor $\gamma$. A policy $\pi(\mathbf{a}|\mathbf{s})$ maps from a state $\mathbf{s}$ to an action $\mathbf{a}$. RL aims to find the optimal policy that maximizes the expected discounted return, $\mathbb{E}_{\mathbf{s}_0 \sim \rho_0, (\mathbf{s}_0, \mathbf{a}_0, ..., \mathbf{s}_{T_i}) \sim \pi} \left[ \sum_{t=0}^{T_i-1} \gamma^t R(\mathbf{s}_t, \mathbf{a}_t) \right]$, where $T_i$ is the variable episode length.

**Unlabeled Offline Data**   We assume access to a reward-free task-agnostic dataset [15, 12], which is a set of $N$ state-action trajectories, $\mathcal{D} = \{\tau_1, \ldots, \tau_N\}$. Since it is task-agnostic, this data can be collected from human teleoperation, unsupervised exploration, or training data for other tasks. We do not assume this dataset contains solutions for the downstream task; therefore, tackling the downstream task requires re-composition of skills learned from diverse trajectories.

**Skill-based RL**   We define skills as a sequence of actions $(\mathbf{a}_0, \ldots, \mathbf{a}_{H-1})$ with a fixed horizon[2] $H$ and parameterize skills as a skill latent $\mathbf{z}$ and skill policy, $\pi^L(\mathbf{a}|\mathbf{s}, \mathbf{z})$, that maps a skill latent and state to the corresponding action sequence. The skill latent and skill policy can be trained using variational

---

[2]It is worth noting that our method is compatible with variable-length skills [26, 39, 27] and goal-conditioned skills [20] with minimal change; however, for simplicity, we adopt fixed-length skills of $H = 10$ in this paper.

auto-encoder (VAE [40]), where a skill encoder $q(\mathbf{z}|(\mathbf{s}, \mathbf{a})_{0:H-1})$ embeds a sequence of transitions into a skill latent $\mathbf{z}$, and the skill policy decodes it back to the original action sequence. Following SPiRL [12], we also learn a skill prior $p(\mathbf{z}|\mathbf{s})$, which is the skill distribution in the offline data, to guide the downstream task policy to explore promising skills over the large skill space.

## 3.2 SkiMo Model Components

*SkiMo* consists of three major model components: the skill policy ($\pi_\theta^L$), skill dynamics model ($D_\psi$), and task policy ($\pi_\phi$), along with auxiliary components for representation learning and value estimation. A state encoder $E_\psi$ first encodes an observation $\mathbf{s}$ into the latent state $\mathbf{h}$. Then, given a skill $\mathbf{z}$, the skill dynamics $D_\psi$ predicts the skill effect in the latent space. The task policy $\pi_\phi$, reward function $R_\phi$, and value function $Q_\phi$ predict a skill, reward, and value on the (imagined) latent state, respectively. The following is a summary of the notations of our model components:

$$
\begin{aligned}
\text{State encoder:} && \mathbf{h}_t &= E_\psi(\mathbf{s}_t) \\
\text{Observation decoder:} && \hat{\mathbf{s}}_t &= O_\theta(\mathbf{h}_t) & \text{Skill dynamics:} && \hat{\mathbf{h}}_{t+H} &= D_\psi(\mathbf{h}_t, \mathbf{z}_t) \\
\text{Skill prior:} && \hat{\mathbf{z}}_t &\sim p_\theta(\mathbf{s}_t) & \text{Task policy:} && \hat{\mathbf{z}}_t &\sim \pi_\phi(\mathbf{h}_t) \\
\text{Skill encoder:} && \mathbf{z}_t &\sim q_\theta((\mathbf{s}, \mathbf{a})_{t:t+H-1}) & \text{Reward:} && \hat{r}_t &= R_\phi(\mathbf{h}_t, \mathbf{z}_t) \\
\text{Skill policy:} && \hat{\mathbf{a}}_t &= \pi_\theta^L(\mathbf{s}_t, \mathbf{z}_t) & \text{Value:} && \hat{v}_t &= Q_\phi(\mathbf{h}_t, \mathbf{z}_t)
\end{aligned} \tag{1}
$$

For convenience, we label the trainable parameters $\psi$, $\theta$, $\phi$ of each component according to which phase they are trained on:

1. **Learned from offline data and finetuned in downstream RL** ($\psi = \{\psi_E, \psi_D\}$): The state encoder ($E_\psi$) and the skill dynamics model ($D_\psi$) are first trained on the offline task-agnostic data and then finetuned in downstream RL to account for unseen states and transitions.

2. **Learned only from offline data** ($\theta = \{\theta_O, \theta_q, \theta_p, \theta_{\pi^L}\}$): The observation decoder ($O_\theta$), skill encoder ($q_\theta$), skill prior ($p_\theta$), and skill policy ($\pi_\theta^L$) are learned from the offline data.

3. **Learned in downstream RL** ($\phi = \{\phi_Q, \phi_R, \phi_\pi\}$): The value ($Q_\phi$) and reward ($R_\phi$) functions, and the task policy ($\pi_\phi$) are trained for the downstream task using environment interactions.

## 3.3 Pre-Training Skill Dynamics Model and Skills from Task-agnostic Data

*SkiMo* consists of pre-training and downstream RL phases. In pre-training, *SkiMo* leverages offline data to extract (1) skills for temporal abstraction of actions, (2) skill dynamics for skill-level planning on a latent state space, and (3) a skill prior [12] to guide exploration. Specifically, we jointly learn a skill policy and skill dynamics model, instead of learning them separately [35, 5, 36], in a self-supervised manner. The key insight is that this joint training could shape the latent skill space $\mathbb{Z}$ and state embedding in that the skill dynamics model can easily predict the future.

In contrast to prior works that learn models completely online [3, 41, 4], we leverage existing offline task-agnostic datasets to pre-train a skill dynamics model and skill policy. This offers the benefit that the model and skills are agnostic to specific tasks so that they may be used in multiple tasks. Afterwards in the downstream RL phase, the agent continues to finetune the skill dynamics model to accommodate task-specific trajectories.

To learn a low-dimensional skill latent space $\mathbb{Z}$ that encodes action sequences, we train a conditional VAE [40, 42] on the offline dataset that reconstructs the action sequence through a skill embedding given a state-action sequence as in SPiRL [12, 16]. Specifically, given $H$ consecutive states and actions $(\mathbf{s}, \mathbf{a})_{0:H-1}$, a skill encoder $q_\theta$ predicts a skill embedding $\mathbf{z}$ and a skill decoder $\pi_\theta^L$ (i.e. the low-level skill policy) reconstructs the original action sequence from $\mathbf{z}$:

$$
\mathcal{L}_{\text{VAE}} = \mathbb{E}_{(\mathbf{s},\mathbf{a})_{0:H-1}\sim\mathcal{D}} \left[ \frac{\lambda_{\text{BC}}}{H} \sum_{i=0}^{H-1} \underbrace{(\pi_\theta^L(\mathbf{s}_i, \mathbf{z}) - \mathbf{a}_i)^2}_{\text{Behavioral cloning}} + \beta \cdot \underbrace{KL\big(q_\theta(\mathbf{z}|(\mathbf{s},\mathbf{a})_{0:H-1}) \,\|\, p(\mathbf{z})\big)}_{\text{Embedding regularization}} \right], \tag{2}
$$

where $\mathbf{z}$ is sampled from $q_\theta$ and $\lambda_{\text{BC}}, \beta$ are weighting factors for regularizing the skill latent $\mathbf{z}$ distribution to a prior of a $\tanh$-transformed unit Gaussian distribution, $Z \sim \tanh(\mathcal{N}(0,1))$.

To ensure the latent skill space is suited for long-term prediction, we *jointly* train a skill dynamics model with the VAE above. The skill dynamics model learns to predict $\mathbf{h}_{t+H}$, the latent state $H$-steps

ahead conditioned on a skill $\mathbf{z}$, for $N$ sequential skill transitions using the latent state consistency loss [4]. To prevent a trivial solution and encode rich information from observations, we additionally train an observation decoder $O_\theta$ using the observation reconstruction loss. Altogether, the skill dynamics $D_\psi$, state encoder $E_\psi$, and observation decoder $O_\theta$ are trained on the following objective:

$$\mathcal{L}_{\text{REC}} = \mathbb{E}_{(\mathbf{s},\mathbf{a})_{0:NH}\sim\mathcal{D}}\left[\sum_{i=0}^{N-1}\left[\underbrace{\lambda_{\text{O}}\|\mathbf{s}_{iH} - O_\theta(E_\psi(\mathbf{s}_{iH}))\|_2^2}_{\text{Observation reconstruction}} + \underbrace{\lambda_{\text{L}}\|D_\psi(\hat{\mathbf{h}}_{iH},\mathbf{z}_{iH}) - E_{\psi^-}(\mathbf{s}_{(i+1)H})\|_2^2}_{\text{Latent state consistency}}\right]\right],$$

(3)

where $\lambda_{\text{O}}, \lambda_{\text{L}}$ are weighting factors and $\hat{\mathbf{h}}_0 = E_\psi(\mathbf{s}_0)$ and $\hat{\mathbf{h}}_{(i+1)H} = D_\psi(\hat{\mathbf{h}}_{iH},\mathbf{z}_{iH})$ such that gradients are back-propagated through time. For stable training, we use a target network whose parameter $\psi^-$ is slowly soft-copied from $\psi$.

Furthermore, to guide the exploration for downstream RL, we also extract a skill prior [12] from offline data that predicts the skill distribution for any state. The skill prior is trained by minimizing the KL divergence between output distributions of the skill encoder $q_\theta$ and the skill prior $p_\theta$:

$$\mathcal{L}_{\text{SP}} = \mathbb{E}_{(\mathbf{s},\mathbf{a})_{0:H-1}\sim\mathcal{D}}\left[\lambda_{\text{SP}}\cdot KL\Big(\mathbf{sg}(q_\theta(\mathbf{z}|\mathbf{s}_{0:H-1},\mathbf{a}_{0:H-1}))\,\|\,p_\theta(\mathbf{z}|\mathbf{s}_0)\Big)\right],$$

(4)

where $\lambda_{\text{SP}}$ is a weighting factor and $\mathbf{sg}$ denotes the stop gradient operator.

Combining the objectives above, we jointly train the policy, model, and prior, which leads to a well-shaped skill latent space that is optimized for both skill reconstruction and long-term prediction:

$$\mathcal{L} = \mathcal{L}_{\text{VAE}} + \mathcal{L}_{\text{REC}} + \mathcal{L}_{\text{SP}}$$

(5)

### 3.4 Downstream Task Learning with Learned Skill Dynamics Model

To accelerate downstream RL with the learned skill repertoire, *SkiMo* learns a high-level task policy $\pi_\phi(\mathbf{z}_t|\mathbf{h}_t)$ that outputs a latent skill embedding $\mathbf{z}_t$, which is then translated into a sequence of $H$ actions using the pre-trained skill policy $\pi_\theta^L$ to act in the environment [12, 16].

To further improve the sample efficiency, we propose to use model-based RL in the skill space by leveraging the skill dynamics model. The skill dynamics model and task policy can generate imaginary rollouts in the skill space by repeating (1) sampling a skill, $\mathbf{z}_t \sim \pi_\phi(\mathbf{h}_t)$, and (2) predicting $H$-step future after executing the skill, $\mathbf{h}_{t+H} = D_\psi(\mathbf{h}_t,\mathbf{z}_t)$. Our skill dynamics model requires only $1/H$ dynamics predictions and action selections of the flat model-based RL approaches [3, 4], resulting in more efficient and accurate long-horizon imaginary rollouts (see Appendix, Figure 10).

Following TD-MPC [4], we leverage these imaginary rollouts both for planning (Algorithm 2) and policy optimization (Equation (7)), significantly reducing the number of necessary environment interactions. During rollout, we perform Model Predictive Control (MPC), which re-plans every step using CEM and executes the first skill of the skill plan (see Appendix, Section C for more details).

To evaluate imaginary rollouts, we train a reward function $R_\phi(\mathbf{h}_t,\mathbf{z}_t)$ that predicts the sum of $H$-step rewards[3], $r_t$, and a Q-value function $Q_\phi(\mathbf{h}_t,\mathbf{z}_t)$. We also finetune the skill dynamics model $D_\psi$ and state encoder $E_\psi$ on the downstream task to improve the model prediction:

$$\mathcal{L}'_{REC} = \mathbb{E}_{\mathbf{s}_t,\mathbf{z}_t,\mathbf{s}_{t+H},r_t\sim\mathcal{D}}\left[\underbrace{\lambda_{\text{L}}\|D_\psi(\hat{\mathbf{h}}_t,\mathbf{z}_t) - E_{\psi^-}(\mathbf{s}_{t+H})\|_2^2}_{\text{Latent state consistency}} + \underbrace{\lambda_{\text{R}}\|r_t - R_\phi(\hat{\mathbf{h}}_t,\mathbf{z}_t)\|_2^2}_{\text{Reward prediction}}\right.$$

$$\left. + \underbrace{\lambda_{\text{V}}\|r_t + \gamma Q_{\phi^-}(\hat{\mathbf{h}}_{t+H},\pi_\phi(\hat{\mathbf{h}}_{t+H})) - Q_\phi(\hat{\mathbf{h}}_t,\mathbf{z}_t)\|_2^2}_{\text{Value prediction}}\right].$$

(6)

Finally, we train a high-level task policy $\pi_\phi$ to maximize the estimated Q-value while regularizing it to the pre-trained skill prior $p_\theta$ [12], which helps the policy output plausible skills:

$$\mathcal{L}_{RL} = \mathbb{E}_{\mathbf{s}_t\sim\mathcal{D}}\big[-Q_\phi(\hat{\mathbf{h}}_t,\pi_\phi(\mathbf{sg}(\hat{\mathbf{h}}_t))) + \alpha\cdot KL\big(\pi_\phi(\mathbf{z}_t|\mathbf{sg}(\hat{\mathbf{h}}_t))\,\|\,p_\theta(\mathbf{z}_t|\mathbf{s}_t)\big)\big].$$

(7)

The models $(\psi,\phi)$ are trained to minimize Equation (6) and Equation (7) using back-propagation through time over $N$ consecutive skill-level transitions, similar to Equation (3).

---

[3]For clarity, we use $r_t$ to abbreviate the sum of $H$-step environment rewards $\sum_{i=0}^{H-1} R(\mathbf{s}_{t+i},\mathbf{a}_{t+i})$.

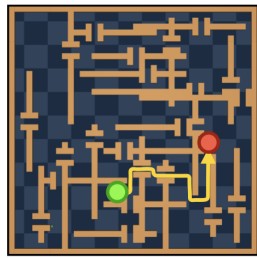 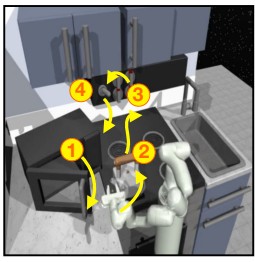 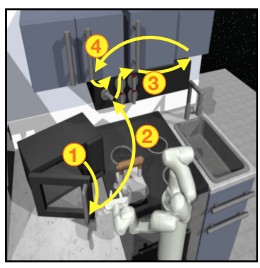 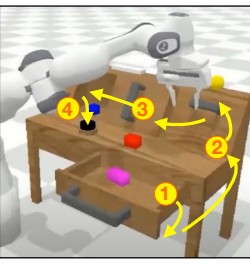

| (a) Maze | (b) Kitchen | (c) Mis-aligned Kitchen | (d) CALVIN |

Figure 3: We evaluate our method on four long-horizon, sparse reward tasks. (a) The green point mass navigates the maze to reach the goal (red). (b, c) The robot arm in the kitchen must complete four tasks in the correct order, *Microwave - Kettle - Bottom Burner - Light* and *Microwave - Light - Slide Cabinet - Hinge Cabinet*. (d) The robot arm needs to complete four tasks in the correct order, *Open Drawer - Turn on Lightbulb - Move Slider Left - Turn on LED*.

## 4 Experiments

In this paper, we propose *SkiMo*, a model-based RL approach that can efficiently and accurately plan long-horizon trajectories by leveraging skills and skill dynamics model. In our experiments, we aim to answer the following questions: (1) Can the skill dynamics model improve the efficiency of RL for long-horizon tasks? and (2) Is the joint training of skills and the skill dynamics model essential for efficient model-based RL?

We compare SkiMo with prior model-based RL and skill-based RL methods on four long-horizon tasks with sparse rewards: maze navigation, two kitchen manipulation, and tabletop manipulation tasks, as illustrated in Figure 3. More experimental details can be found in Appendix, Section C.

### 4.1 Tasks

**Maze**  We use the maze navigation task from Pertsch et al. [16], where a point mass agent is randomly initialized near the green region and needs to reach the fixed goal region in red (Figure 3a). The agent observes its 2D position and 2D velocity, and controls its $(x, y)$-velocity. The agent receives a sparse reward of 100 only when it reaches the goal. The task-agnostic dataset [16] consists of 3,046 trajectories between randomly sampled initial and goal positions.

**Kitchen**  We use the FrankaKitchen tasks and 603 trajectories from D4RL [43]. The 7-DoF Franka Emika Panda arm needs to perform four sequential sub-tasks, *Microwave - Kettle - Bottom Burner - Light*. In **Mis-aligned Kitchen**, we also test another task sequence, *Microwave - Light - Slide Cabinet - Hinge Cabinet*, which has a low sub-task transition probability in the offline data distribution [16]. The agent observes 11D robot state and 19D object state, and uses 9D joint velocity control. The agent receives a reward of 1 for every sub-task completion in order.

**CALVIN**  We adapt CALVIN [44] to have the target task, *Open Drawer - Turn on Lightbulb - Move Slider Left - Turn on LED*, and 21D robot and object states. It uses the Franka Panda arm with 7D end-effector pose control. The offline data is from play data [44] consisting of 1,239 trajectories. The agent receives a reward of 1 for every sub-task completion in the correct order.

### 4.2 Baselines

- **Dreamer** [3] and **TD-MPC** [4] learn a flat (single-step) dynamics and train a policy using latent imagination to achieve a high sample efficiency.
- **DADS** [29] discovers skills and learns a dynamics model through unsupervised learning.
- **LSP** [36] plans in the skill space, but using a single-step dynamics model from Dreamer [3].
- **SPiRL** [12] learns skills and a skill prior, and guides a high-level policy using the learned prior.
- **SPiRL + Dreamer** and **SPiRL + TD-MPC** pre-train the skills using SPiRL and learn a policy and model in the skill space, instead of the low-level action space, using Dreamer and TD-MPC, respectively. In contrast to *SkiMo*, these baselines do not jointly train the model and skills.

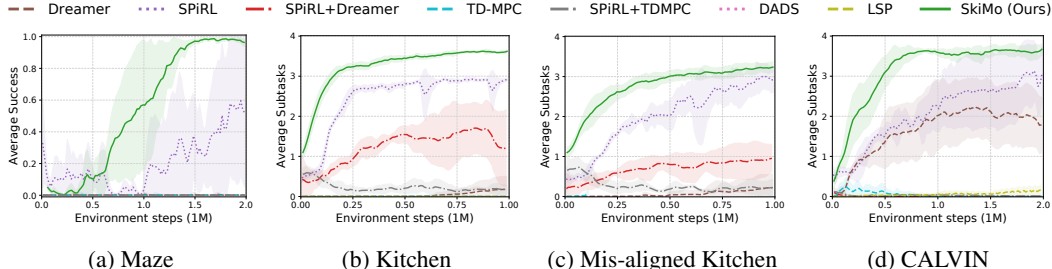

Figure 4: Learning curves of our method and baselines. All averaged over 5 random seeds.

## 4.3 Results

**Maze** Maze navigation poses a hard exploration problem due to the sparsity of the reward: the agent only receives reward after taking 1,000+ steps to reach the goal. Figure 4a shows that only *SkiMo* is able to consistently reach the goal, whereas baselines struggle to learn a policy or an accurate model due to the challenges in sparse feedback and long-term planning.

We qualitatively analyze the behavior of each agent in Appendix, Figure 9. Dreamer and TD-MPC have a small coverage of the maze since it is challenging to coherently explore for 1,000+ steps to reach the goal from taking primitive actions. Similarly, DADS and LSP could not learn meaningful skills and never find the goal. SPiRL is able to explore a large fraction of the maze, but it does not learn to consistently find the goal due to difficult policy optimization in long-horizon tasks. SPiRL + Dreamer and SPiRL + TD-MPC fail to learn an accurate model and often collide with walls.

**Kitchen** Figure 4b demonstrates that *SkiMo* reaches the same performance (above 3 sub-tasks) with 5x less environment interactions than SPiRL. In contrast, Dreamer, TD-MPC, DADS, and LSP rarely succeed on the first sub-task due to the sparse reward. SPiRL + Dreamer and SPiRL + TD-MPC perform better than flat model-based RL by leveraging skills, yet the independently trained model and policy are not accurate enough to consistently achieve more than two sub-tasks.

**Mis-aligned Kitchen** The mis-aligned target task makes the downstream learning harder because the skill prior, which reflects offline data distribution, offers less meaningful regularization to the policy. However, Figure 4c shows that *SkiMo* still performs well. This demonstrates that the skill dynamics model is able to adapt to the new distribution of behaviors, which might significantly deviate from the distribution in the offline dataset.

**CALVIN** One of the major challenges in CALVIN is that the offline data is very task-agnostic: any particular sub-task transition has probability lower than 0.1% on average, resulting in a large number of plausible skills from any state. Figure 4d shows that *SkiMo* can learn faster than the model-free baseline, SPiRL, which supports the benefit of using our skill dynamics model. Meanwhile, Dreamer performs better in CALVIN than in Kitchen because objects in CALVIN are more compactly located and easier to manipulate with the end-effector control; thus, it becomes viable to accomplish initial sub-tasks through random exploration. However, it falls short in composing coherent action sequences to achieve a longer task sequence due to the lack of temporally-extended reasoning.

In summary, we show the synergistic benefit of temporal abstraction in both the policy and dynamics model. *SkiMo* is the only method that consistently solves the long-horizon tasks. Our results also demonstrate the importance of algorithmic design choices (e.g. skill-level planning, joint training of a model and skills) as naive combinations (SPiRL + Dreamer, SPiRL + TD-MPC) fail to learn.

## 4.4 Ablation Studies

**Model-based vs. Model-free** In Figure 5, *SkiMo* achieves better asymptotic performance and higher sample efficiency across all tasks than *SkiMo + SAC*, which uses model-free RL (SAC [45]) to train the high-level policy to select skills. The comparison suggests that the task policy can make more informative decisions by leveraging accurate long-term predictions of the skill dynamics model.

**Joint training of skills and skill dynamics model** *SkiMo w/o joint training* learns the latent skill space using only the VAE loss in Equation (2). Figure 5 shows that the joint training is crucial in

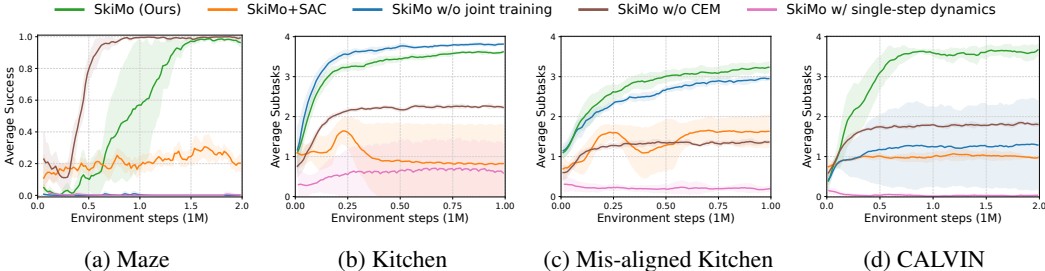

| (a) Maze | (b) Kitchen | (c) Mis-aligned Kitchen | (d) CALVIN |

Figure 5: Learning curves of our method and ablated methods. All averaged over 5 random seeds.

more challenging scenarios, where the agent needs to generate accurate long-term plans (for Maze) or the skills are very diverse (in CALVIN).

**CEM planning**    As shown in Figure 5, *SkiMo* learns significantly better and faster in Kitchen, Mis-aligned Kitchen, and CALVIN than *SkiMo w/o CEM*, indicating that CEM planning can effectively find a better plan. On the other hand, in Maze, *SkiMo w/o CEM* learns twice as fast. We find that action noise for exploration in CEM leads the agent to get stuck at walls and corners. We believe that with a careful tuning of action noise, *SkiMo* can solve Maze much more efficiently.

**Skill dynamics model**    *SkiMo w/ single-step dynamics* replaces the skill dynamics model with a conventional single-step dynamics model. Similar to LSP, it learns and plans on skills, but additionally pre-trains the skills and the model on offline datasets for fair comparison. As shown in Figure 5, the single-step model struggles at handling long-horizon planning on all tasks, akin to the baseline results on DADS and LSP in Figure 4. In contrast, the skill dynamics model can make accurate long-horizon predictions for planning due to significantly less compounding errors.

For further ablations and discussion on skill horizon and planning horizon, see Appendix, Section A.

### 4.5   Long-horizon Prediction with Skill Dynamics Model

To assess the accuracy of long-term prediction of our proposed skill dynamics over flat dynamics, we visualize imagined trajectories in Appendix, Figure 10, where the ground truth initial state and a sequence of 500 actions (50 skills for *SkiMo*) are given. Dreamer struggles to make accurate long-horizon predictions due to error accumulation. In contrast, *SkiMo* is able to reproduce the ground truth trajectory with little prediction error even when traversing through hallways and doorways. This confirms that *SkiMo* allows temporal abstraction in the dynamics model, thereby enabling temporally-extended prediction and reducing step-by-step prediction error.

## 5   Conclusion

We propose *SkiMo*, an intuitive instantiation of saltatory model-based hierarchical RL [2], which combines skill-based and model-based RL approaches. Our experiments demonstrate that (1) a skill dynamics model reduces the long-term future prediction error via temporal abstraction in the dynamics model; (2) without needing to plan step-by-step, downstream RL over the skill space allows for efficient and accurate temporally-extended reasoning, improving the performance of prior model-based RL and skill-based RL; and (3) joint training of the skill dynamics and skills further improves the sample efficiency by learning skills conducive to predict their consequences. We believe that the ability to learn and utilize a skill-level model holds the key to unlocking the sample efficiency and widespread use of RL agents for long-horizon tasks, and our method takes a step toward this direction.

**Limitations and future work**    While our method extracts fixed-length skills from offline data, the lengths of semantic skills may vary based on the contexts and goals. Future work can learn variable-length semantic skills to improve long-term prediction and planning. Further, although we only experimented on state-based inputs, *SkiMo* is a general framework that can be extended to RGB, depth, and tactile observations. Thus, extending our approach to real robots with high-dimensional observations would be an interesting future work.

**Acknowledgments**

This work was supported by Institute of Information & communications Technology Planning & Evaluation (IITP) grants (No.2019-0-00075, Artificial Intelligence Graduate School Program, KAIST; No.2022-0-00077, AI Technology Development for Commonsense Extraction, Reasoning, and Inference from Heterogeneous Data) and National Research Foundation of Korea (NRF) grant (NRF-2021H1D3A2A03103683), funded by the Korea government (MSIT). This work was also partly supported by the Annenberg Fellowship from USC. We would like to thank Ayush Jain and Grace Zhang for help on writing, Karl Pertsch for assistance in setting up SPiRL and CALVIN, Kevin Xie for providing code of LSP, and all members of the USC CLVR lab for constructive feedback.

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
