# OpenReview forum: "Skill-based Model-based Reinforcement Learning"
_robot-learning.org/CoRL/2022/Conference — CoRL 2022 Poster_

### Official Review · Reviewer_gSBE · 2022-07-27

**Originality:** Excellent
**Technical Quality:** Very Good
**Clarity Of Presentation:** Very Good
**Impact:** 4

**Recommendation:**

Weak Accept: I recommend accepting the paper, but will not argue for my recommendation if the majority of other reviewers have a different opinion.

**Summary:**

This paper presents Skill-based Model-based RL framework to efficiently learn downstream policies by jointly training a skill dynamics model and skill encoding networks from task-agnostic data. The skill dynamics model learns temporal consistency by predicting the final state after performing a skill (a sequence of actions) from a given state. Skills are encoded in a latent skill space, the skill dynamics model is used to plan in this latent space using cross-entropy methods and skills sampled from a task-level policy. Trajectories from the dynamics model rollouts are evaluated with learned reward and Q functions, with the best trajectories decoded into action sequences that are executed in the environment. Additional fine-tuning of the skill dynamics model is performed during downstream task learning. The framework is demonstrated in simulation with a maze task and a manipulation task, performing significantly better than separate model-based and skill-based methods.



**Issues:**

Is the negative expectation incorrect in equation 4 (negative expectation), minimising this loss will increase the KL divergence? This seems to be the opposite of equation 2.

Target networks were used through the Algorithms in the Appendix, but were not introduced in the paper. This is not really an issue if this detail was omitted for clarity, except for the superscript “-” on the model parameters for Q and E that was not explained anywhere.

Modules were given different names throughout, for example, O is introduced as the reconstruction module, though later referred to as the observation, and as the state decoder. Similar for E. Also, what is “sg” (equation 4 and 7)?

Figure 2: “Prior” should be changed to “Skill Prior” to differentiate this from the fixed-prior used to regularise the skill encoder / decoder. q(z|s0,a) takes a vector of actions? Maybe make “a” bold to highlight this. No model actually takes “s” as input except the skill encoder, is that correct? Instead, “h” is used from the state encoder. It makes sense this is omitted from the diagram for clarity, but should probably be mentioned in the caption or where the figure is introduced in the text.

Equation 7, should be pθ, not p_phi to regularise the skill prior pθ? Also, should be π_phi not πθ to regularise the policy, not the skill decoder?

Minor:
Figure 1: ”Skill dynamics model with skills” maybe just ”Skill dynamics model” is sufficient.

Minor:
There were two mentions of furniture assembly as a complex long horizon task. I was expecting this as an experiment with improvements on the millions to billions of interactions. Perhaps include that maze navigation and kitchen tasks are also very complex long horizon tasks.

Spelling / Grammar / Typos

“first work that jointly learning”

“for more sample efficiency policy learning”

“Conventionally model-based approaches” -> missing comma, or change to “Conventional”

“SkiMo is the first work that jointly learning skills”

“high level policy” or “high-level policy”, both used throughout.

“crucial in the real world robot learning”


**Quality Of The Limitations Section:**

Limitations are addressed clearly

**Reviewer Expertise:**

4: The reviewer is confident but not absolutely certain that the evaluation is correct

**Robotics Focus:**

Highly relevant to robotics but no hardware experiments

**Strengths And Weaknesses:**

This paper presents a combination of techniques that improve on model-based and skill-based methods. The ideas and motivations are expressed clearly and the paper is well written. The comparisons appear relevant and fair, the results show a clear benefit of the presented method. The ablation study highlighted the benefit of the skill dynamics model and of joint training. The extensive appendix was welcomed and the additional experiments that didn’t make it into the paper shone further light on the method and the improvements over existing approaches.

The primary weakness (apart from the lack of real robot experiments) was confusion through the method section, particularly Section 3.2.1 where components were introduced, but not defined until the following section (3.2.2). For example, there is no mention of “h” before equation 1. It might be better to move some definitions into 3.2.1, or re-work these two sections. Also consider re-ordering the list in Section 3.2.1 to follow the flow of learning: first train on task-agnostic data, then fine-tune / train on downstream tasks.

Questions:

Was there a benefit using the state representation and observation modules? Were these modules used in the baselines? Is there utility in training these with target networks?

Was the maze navigation task different in this work compared with the SPiRL paper? The performance of SPiRL seems much worse than the original paper, is this due to the original paper using more goal–reaching trajectories in their training data?

What is the reason success on all four sequential tasks in the Kitchen environment is difficult? Do the authors have an intuition about how to more robustly solve longer task sequences?

What is the effect of fine-tuning after training? Is this expected to be a significant component for real-world experiments?


**Summary Of Recommendation:**

The recommendation is to improve the clarity of the method section.

---

> ### Author Response · Authors · 2022-08-19
> **Response to Reviewer gSBE**
>
> Thank you for your constructive feedback. We address your concerns in detail below and have updated our paper accordingly.
>
> &nbsp;
>
>
> **Confusion in Section 3.2.1**
>
> Thank you for the great suggestions! We have updated Section 3.1 and 3.2 accordingly to improve the clarity of Section 3.2.1.
>
> &nbsp;
>
>
> **Was there a benefit using the state representation and observation modules? Were these modules used in the baselines? Is there utility in training these with target networks?**
>
> We use the state representation module (state encoder) so that our framework can be easily extended to any form of high-dimensional inputs such as RGB, depth, and tactile observations. The observation module (observation decoder) is used to provide the auxiliary observation reconstruction loss to prevent collapsing of the latent state (L151-152).
>
> Both modules are used in Dreamer. Target networks in TD-MPC are adopted for stable training of the latent state space.
>
> &nbsp;
>
>
> **Was the maze navigation task different in this work compared with the SPiRL paper?**
>
> Yes, it is different from the one in SPiRL. The maze navigation task we used is from a follow-up work to SPiRL, SkiLD [a], where the task-agnostic data is collected on the 40x40 maze (same as the testing maze) rather than 20x20. Also at test time, the agent has a random start location instead of a fixed one, which makes the maze navigation task more challenging.
>
> &nbsp;
>
>
> **What is the reason success on all four sequential tasks in the Kitchen environment is difficult? Do the authors have an intuition about how to more robustly solve longer task sequences?**
>
> This environment is difficult for RL because of the sparse reward and the high DoF action space, which makes random exploration and correct behavior sequencing hard to learn.
>
> To robustly solve longer task sequences, we hypothesize that it requires hierarchical planning and generalizable skills. The intuition is that humans, instead of planning every muscle movement like MBRL today, conceive action plans at multiple levels of abstraction. We think this would be an interesting future avenue of research, and our work takes a step towards this direction.
>
> &nbsp;
>
>
> **What is the effect of fine-tuning after pre-training? Is this expected to be a significant component for real-world experiments?**
>
> Great question! With fine-tuning, our agent can adapt to new tasks and environment states unseen in the offline task-agnostic data. We think this is worth studying, so we compared the performance on fine-tuning and freezing the skill dynamics model during downstream RL. We found that without fine-tuning, the agent performs much worse due to the discrepancy between distributions of offline data and the downstream task. We have included these results in Appendix, A.3.
>
> Fine-tuning the skill dynamics model can be especially important for real-world experiments as robots will likely encounter new observations and dynamics in the real-world environment.
>
> &nbsp;
>
>
> **Is the negative expectation incorrect in Equation 4?**
>
> Exactly! We fixed it in the revised version.
>
> &nbsp;
>
>
> **Target networks are not introduced in the paper.**
>
> We revised the paper accordingly (now introduced after Equation 4).
>
> &nbsp;
>
>
> **Modules were given different names throughout.**
>
> Thank you for pointing this out! We have made the names consistent in the revised version: state encoder $E$ and observation decoder $O$.
>
> &nbsp;
>
>
> **What is sg?**
>
> Thank you for spotting this. “sg” denotes the “stop gradients” operator, which does not back-propagate gradients to the skill encoder $q_\theta$ in Equation 4 and the state encoder $h_t=E_\phi(s_t)$ in Equation 7. We added the definition of “sg” after Equation 4.
>
> &nbsp;
>
>
> **In Figure 2, “Prior” should be changed to “Skill Prior”.**
>
> Thank you for the suggestion. We agree that “Skill Prior” is more precise, and updated Figure 2 accordingly.
>
> &nbsp;
>
>
> **In Figure 2, $q(z \vert s_0, a) takes a tuple of actions?**
>
> Thank you for your suggestion. The skill encoder takes a tuple of states and actions, so we bold both states and actions in the revised paper.
>
> &nbsp;
>
>
> **No model actually takes $s$ as input.**
>
> Thank you for pointing this out. We added clarification in the caption of Figure 2.
>
> &nbsp;
>
>
> **In Equation 7, $p_\phi$ should be $p_\theta$ and $\pi_\theta$ should be $\pi_\phi$.**
>
> You are right. We corrected Equation 7 accordingly.
>
> &nbsp;
>
>
> **Typos**
>
> Thank you for spotting the typos and grammatical errors. We fixed all of them.
>
>
> &nbsp;
>
>
> **References**
>
> [a] Pertsch et al. Demonstration-Guided Reinforcement Learning with Learned Skills. CoRL 2021.

---

> > ### Comment · Reviewer_gSBE · 2022-08-25
> > **Response to authors**
> >
> > Thank you for the detailed response, I have no further suggestions and think these contributions are valuable for the community.

---

### Official Review · Reviewer_43gq · 2022-07-28

**Originality:** Fair
**Technical Quality:** Good
**Clarity Of Presentation:** Fair
**Impact:** 3

**Recommendation:**

Weak Reject: I recommend rejecting the paper, but will not argue for my recommendation if the majority of other reviewers have a different opinion.

**Summary:**

The work proposes to combine option (skill)-based learning with model-based RL, to achieve a more sample efficient and robust (higher probability of success) approach. The proposed approach relies on demonstrations to pre-train several model components: the state representation module (E_\psi) and the skill dynamics module (D_\psi), as well as the state decoder (O_\theta) in order to avoid trivial solutions. These pre-trained components are then used in the second phase of training which involves fine tuning them on a rewards-prediction/latent-state prediction, and finally an RL based objective that performs model-predictive control using a recently proposed method, TD-MPC.  Experiments on the maze and FrankaKitchen suites from D4RL demonstrates the method's superior performance. Ablation studies demonstrate the utility of jointly training the skill-representation component & skill-dynamics model.

**Issues:**

1-Do E & D share parameters? If not, then subscripting both with \psi is very confusing.
2-Reconstructing the observation space is known to be problematic, i.e. there are simple solutions that do not capture the details relevant to the problem. Can you give more details on this in the context of your proposed approach?
3-In Equation 2 is z in the first term output by q_\theta (of the second term)?
4-Also Equation 2. Why do you apply tanh to the Gaussian samples, i.e. why is it important to clamp their values to be in [-1, 1]?
5-Equation 3: is E_{\psi^-} a typo (last term)? Or is it some sort of out-dated parameter vector used in the loss?
6-How is the horizon length (skill vector temporal length) found?
7-What is "sg" in Equation 4?
8-Are there no multipliers on the various loss terms in Equation 6? i.e. are they all 1? If so, why? If not, how are the hparams tuned (in all loss terms)?

**Quality Of The Limitations Section:**

Additional details required

**Reviewer Expertise:**

4: The reviewer is confident but not absolutely certain that the evaluation is correct

**Robotics Focus:**

Highly relevant to robotics but no hardware experiments

**Strengths And Weaknesses:**

Strengths:
1-The paper's language is pretty clear and it is well written
2-The performance gains seem impressive on the tasks presented (although I am not familiar with the performance of related works)

Weaknesses:
1-The method seems *extremely* complicated and would be difficult to reproduce in my opinion. For example, the total number of terms to minimize in all objective functions is 10, and the total number of "model components" (i.e. networks) is 9 (!!!).
2-The method omits the "Androit" and Gym suite of tasks from D4RL although they seem to be well suited (i.e. they all include demonstrations). Why not include these tasks as well?
3-Many implementation details are missing, not least the hpram details.
4-Despite the paper being generally well written, some technical details require major clarifications.

**Summary Of Recommendation:**

Despite the paper presenting good performance on 2 our of the 4 D4RL suites, I recommend rejection of the paper in the current form.
The main reasons being: 1-the complexity of the proposed approach, and lack of technical details, makes it unreproducible in my opinion -- and therefore of limited value to the robotics/learning community. 2-The proposed model was only demonstrated on two tasks. Neither of which included real robots. This makes me question the robustness and generality of the proposed approach.

---

> ### Author Response · Authors · 2022-08-19
> **Response to Reviewer 43gq (1/2)**
>
> Thank you for your constructive feedback. We address your concerns in detail below and have updated our paper accordingly.
>
> &nbsp;
>
>
> **The complexity of the proposed approach and lack of technical details make it unreproducible.**
>
> This is a valid concern. We appreciate that your comments have helped us improve the clarity of our method section in the revised version! We tried to include most of the important details in the main paper; but, due to the limited space, we left some details in the supplementary materials (the appendix and code). Please let us know if there are any missing details that prevent you from reproducing our method!
>
> Above all, although our method looks complicated at a glance, its implementation leverages a simple combination of well-explored approaches, such as Dreamer, SPiRL, and CEM. In other words, each component of our method has been widely verified and thus, easy to get reproduced.
>
> Furthermore, 4 out of 9 components (observation decoder, skill encoder, skill prior, skill policy) are **entirely pre-trained using supervised learning**, which is easy to train and is done offline. The online RL training phase matches the complexity of typical model-based RL (e.g. Dreamer and TD-MPC), which trains only 5 **essential** components (state encoder, dynamics model, reward function, value function, and task policy).
>
> Finally, we believe that the number of components and loss terms are not metrics to measure whether the method is valuable or whether it is reproducible. We also provided the code and data as supplementary materials for reproducing the results.
>
> &nbsp;
>
>
> **The proposed model was only demonstrated on two tasks. The experiments omit the Adroit and gym tasks from D4RL.**
>
> Our focus is on solving **novel long-horizon tasks**, whereas Adroit and other Gym tasks consist of short-horizon or single-skill (i.e. demonstrated) tasks, which is not suitable for our purpose.
>
> Nevertheless, we highlight that we had **two additional tasks** in the supplementary materials (now merged into the main paper). These additional experiments demonstrate consistent results with the ones in the main paper, reassuring us about our method’s **robustness** and **generality**.
>
> Altogether, we evaluate our method on **four** long-horizon, sparse-reward tasks that cover challenges in exploration, skill composition, generalization, and extremely task-agnostic datasets. Our method shows significant improvement in both performance and sample efficiency over prior works.
>
> &nbsp;
>
>
> **Many implementation details are missing. No hyperparameter details.**
>
> Due to the limited space, we included most implementation details in the supplementary materials. The hyperparameters are exhaustively listed in Appendix, Table 2. In addition, our code can be found in the supplementary materials for reproducibility.
>
> &nbsp;
>
>
> **Despite the paper being generally well written, some technical details require major clarifications.**
>
> Thank you for spotting many typos and ambiguities in our method section! We have revised the entire method section to make it clear and address all your concerns. Your feedback helps us significantly improve our presentation! Please let us know if you find anything unclear.

---

> > ### Author Response · Authors · 2022-08-19
> > **Response to Reviewer 43gq (2/2)**
> >
> > **Do $E$ and $D$ share parameters? If not, subscripting both with $\psi$ is very confusing.**
> >
> > This is a good point! $E$ and $D$ are separate networks with parameters $\psi_E$ and $\psi_D$, respectively, and thus do not share parameters. We apologize for the confusion and we have updated Section 3.2 to clarify it.
> >
> > &nbsp;
> >
> >
> > **Reconstructing the observation space is known to be problematic.**
> >
> > This is another good point! However, our focus is not on representation learning; we use the auxiliary observation reconstruction loss only for preventing the collapse of the latent state during pre-training (L151-152). This can be replaced with other representation learning techniques and we expect more robust representation learning techniques to further enhance our method’s performance.
> >
> > &nbsp;
> >
> >
> > **In Equation 2, is $z$ in the first term output by $q_\theta$?**
> >
> > Yes.
> >
> > &nbsp;
> >
> >
> > **In Equation 2, why do you apply $\tanh$ to the Gaussian samples?**
> >
> > We use the tanh-transformed Gaussian to have a bounded action space $[-1,1]$ (i.e. skill space) for the high-level policy.
> >
> > &nbsp;
> >
> >
> > **In Equation 3, is $E_{\psi^-}$ a typo?**
> >
> > Thank you for pointing this out! $\psi^-$ denotes parameters of the target network. For stable training, we use a target network whose parameter $\psi^-$ is slowly soft-copied from $\psi$, following TD-MPC. We explained it in the revised paper (L155-156).
> >
> > &nbsp;
> >
> >
> > **How is the skill length found?**
> >
> > We started with the skill length 10, which is commonly used in prior skill-based RL (e.g. SPiRL). In Appendix, Section A.1, we included an ablation study on it to test different skill lengths, {1,5,10,15,20}. The results show that the skill length 10 works the best in most cases.
> >
> > &nbsp;
> >
> >
> > **What is “sg” in Equation 4?**
> >
> > Thank you for spotting this out. “sg” denotes the “stop gradients” operator, which does not back-propagate gradients to the skill encoder $q_\theta$ in Equation 4 and the state encoder $h_t=E_\phi(s_t)$ in Equation 7. We added the definition of “sg” after Equation 4.
> >
> > &nbsp;
> >
> >
> > **Are there no multipliers on the various loss terms in Equation 6?**
> >
> > Thanks for pointing out the missing weighting factors of loss terms. To make this point clear, we have explicitly specified the weighting factors in Equation 2-6 in the revised version, which are mostly from prior work (Dreamer, TD-MPC, and SPiRL). The full list of hyperparameters can be found in Appendix, Table 2.

---

> > > ### Comment · Reviewer_43gq · 2022-08-25
> > > **Reviewer Response**
> > >
> > > Thank you for your response and updates to the paper. Despite good performance on some tasks, I can't find a core idea that would be of general use to the wider robotics community and therefore can not recommend the paper for publication.

---

> > > > ### Author Response · Authors · 2022-08-27
> > > > **Response to Reviewer 43gq’s Response**
> > > >
> > > > **What is the core idea that would be of general use to the wider robotics community**
> > > >
> > > > One major bottleneck in robot learning is sample efficiency, and to address this problem model-based RL offers a promising solution. However, the state-of-the-art MBRL methods still cannot solve complex, long-horizon robotic tasks.
> > > >
> > > > We pinpoint the important missing ingredient from prior works in MBRL: the notion of temporal abstraction. Planning based on consecutive single-step predictions is not only inefficient but also prone to error accumulation, which is especially problematic when planning over a long horizon for complex tasks. Thus, we digress from this conventional paradigm and offer temporal abstraction in both the skills and dynamics model.
> > > >
> > > > In our experiments, we show that SkiMo (1) decreases error accumulation in long-term future prediction, (2) allows more temporally-extended planning, and (3) improves sample efficiency. We demonstrate that SkiMo consistently outperforms single-step model-based RL, which has recently shown its applicability in real-world robots. We believe our extensive experiments on three complex robotic manipulation tasks suggest its applicability to real-world robots and it will be our future work.
> > > >
> > > > Thanks again for your valuable feedback. Please let us know if this addresses your concern and if there is any further concern that potentially prevents you from accepting this paper.

---

> ### Author Response · Authors · 2022-08-24
> **Follow up on response**
>
> Thank you for your valuable comments and feedback. We hope our rebuttal has addressed all your concerns. Please let us know if you need any further clarification! We would love to answer your questions and make our paper stronger!

---

### Official Review · Reviewer_ctWV · 2022-07-28

**Originality:** Fair
**Technical Quality:** Good
**Clarity Of Presentation:** Good
**Impact:** 2

**Recommendation:**

Weak Reject: I recommend rejecting the paper, but will not argue for my recommendation if the majority of other reviewers have a different opinion.

**Summary:**

This paper proposes an approach for hierarchical model-based reinforcement learning, where the agent first learns a repertoire of skills, and a skill dynamics model from offline dataset, and then trains a hierarchical policy conditioned on the skill space.

The key contributions of the paper are jointly learning a skill prior and a skill dynamics model from offline data, and demonstrating that the a policy conditioned on skills sampled from the skill prior can be used to guide online exploration in downstream tasks. The paper demonstrates results with the proposed algorithm on simulated robot manipulation (Frank Kitchen) and point agent navigation (maze) environments involving long-horizon tasks.

**Issues:**

Kindly explain the issues I listed (1-4) in the Weaknesses section.

**Quality Of The Limitations Section:**

Limitations are not well addressed

**Reviewer Expertise:**

4: The reviewer is confident but not absolutely certain that the evaluation is correct

**Robotics Focus:**

Relevant but unlikely to deploy to hardware in near future

**Strengths And Weaknesses:**

Strengths:

1. The key insight of learning re-usable skills from offline data and using the learned skill distribution to condition policies for downstream tasks is very well motivated, and relevant to a lot of robot learning settings.

2. The paper is well-written. The algorithm design is well motivated, and easy to follow. The related work positions the paper well in the context of prior model based hierarchical RL works. Sufficient details are provided in the experiments to understand the setting and results well.

3. The modifications of the method over prior work (e.g. Dreamer) are explained well, and logically positioned.

Weaknesses:

1. The paper's insights regarding learning a skill space to condition a policy in model-based RL, overlaps with several prior works as mentioned in the related works section [28, 5, 29, 30]. However, The point of distinction mentioned "To fully unleash the potential of temporally abstracted 84 skills, we devise a skill-level dynamics model to provide accurate long-term prediction, which is
85 essential for solving long-horizon tasks." is not convincing to me - I don't see any reason for the skill-space to be enable long-term prediction because although the skills are low-dimensional actions (as learned by a VAE latent embedding), there is no temporal chaining of skills - skills are sampled IID from the same Gaussian distribution at each time-step, and this is infact exactly similar to the prior works cited previously.

2. All the loss terms in equation (5) are weighted equally - it is unclear why this modeling choice makes sense. Since all the terms in equation (5) are ad-hoc and do not correspond to a principled variational lower bound to the overall RL objective, I believe it is necessary to motivate their relative importance, and ablate this choice through experiments.

3. It is unclear how it is ensured that the policy conditioned on the skill actually makes use of the skill and doesn't simply ignore it? There is no loss term in the objective to actually incentivize making the skills chosen predictable from the observed rollouts. In addition, this is not evaluated through the experiments.

4. The experiment details and results are thorough in the settings considered, but several relevant baselines are not evaluated. The baseline Dreamer is simply a flat model-based RL approach (no hierarchy) and all the other baselines are ablations of the approach. I think it is very important to compare with several external hierarchical RL approaches proposed in the literature over the years, for example [A] HIRO and [B] DADS, the latter of which is very relevant in learning a skill-level dynamics model. Also I think [A] should be cited in the related works.

[A] Nachum, Ofir, et al. "Data-efficient hierarchical reinforcement learning." Advances in neural information processing systems 31 (2018).
[B] Sharma, A., Gu, S., Levine, S., Kumar, V., & Hausman, K. (2019). Dynamics-aware unsupervised discovery of skills. arXiv preprint arXiv:1907.01657.




**Summary Of Recommendation:**

I believe the paper targets an important problem in robotics, that of learning re-usable skills from prior data. However, due to the limitations I pointed out in the weaknesses section, I believe the paper is not ready for publication in its current form.

---

> ### Author Response · Authors · 2022-08-19
> **Response to Reviewer ctWV**
>
> Thank you for your constructive feedback. We address your concerns in detail below and have updated our paper accordingly.
>
> &nbsp;
>
>
> **I don’t see any reason for the skill space to enable long-term prediction.**
>
> We would like to clarify that our **skill dynamics model** enables accurate long-term prediction and temporally-extended reasoning.
>
> First, the skill dynamics model directly predicts the effect of a skill and skips the low-level details of skill execution. This leads to **accurate long-term prediction** than conventional single-step models due to less error accumulation when reasoning over long horizons, as illustrated in Figure 1. Appendix, Figure 10 visualizes that the prediction of the skill dynamics model has little error over 500 timesteps while the prediction from the flat model deviates from the ground truth quickly.
>
> Further, we chain skills through **CEM planning** in the skill space using the skill dynamics model. For example, in Maze, the agent would imagine the next 10 skills and evaluate the utility (reward + value) of this plan. This enables long-horizon reasoning, which is absent from prior works in model-based RL and hierarchical RL.
>
> &nbsp;
>
>
> **Comparison to hierarchical RL approaches [a,b]**
>
> We thank the reviewer for suggesting the highly relevant works, DADS [a] and HIRO [b], which propose to efficiently learn complex tasks using hierarchical policies. We have added these papers to related work.
>
> We conducted additional experiments and found that **DADS could not get any reward on FrankaKitchen**. DADS learns diverse behaviors but they do not contain meaningful skills. Similarly, Figure 5a in Gupta et al. [c] shows that **HIRO fails to learn FrankaKitchen** (i.e. gets near 0 reward).
>
> We would like to emphasize that we instead compared our method against a strictly stronger hierarchical RL method, SPiRL [d], which leverages large offline data, whereas DADS and HIRO learn solely online.
>
> &nbsp;
>
>
> **Distinction with prior model-based RL that uses skills**
>
> We have added more discussion about prior model-based RL with skills to related work in the revised version.
> * Prior model-based RL approaches [a,e,f] also plan over skills. However, their dynamics models make **single-step predictions** (conditioned on the skill latent) as conventional model-based RL, which struggles at handling long-horizon planning due to error accumulation.
> * Wu et al. [g] proposes to learn a temporally-extended dynamics model; however, it conditions on low-level actions rather than skills and is only used for low-level planning.
> * A concurrent work, Shah et al. [h], is most similar to our paper in that it learns a skill dynamics model, but it uses a limited set of discrete, manually-defined skills.
>
> Instead, we devise a skill-level dynamics model that learns to make accurate long-term predictions with a jointly extracted skill library for scalable long-horizon learning.
>
> &nbsp;
>
>
> **Weighting of different loss terms**
>
> Thanks for pointing out the missing weighting factors of loss terms. To make this point clear, we explicitly wrote the weighting factors in Equation 2-6 in the revised version.
>
> We simply take the weighting factors from prior work (SPiRL, Dreamer, and TD-MPC) and they work well across four different navigation/manipulation tasks. The full list of hyperparameters can be found in Appendix, Table 2.
>
> &nbsp;
>
>
> **How to ensure that the policy conditioned on the skill does not ignore the skill?**
>
> We assume the offline task-agnostic data contains diverse behaviors. This multi-modality of data enforces a low-level skill policy not to ignore the skill latent $z$ to diversify its behaviors given a state. Otherwise, the policy cannot reconstruct the correct behavior among diverse trajectories in the dataset, which results in a **high reconstruction loss** in VAE training.
>
> More technically, we trained the skill policy using $\beta$-VAE, where $\beta$ controls how much the skill latent affects the policy. Thus, we can ensure that the policy utilizes the skill latent with a small enough $\beta$.
>
> &nbsp;
>
>
> **References**
>
> [a] Sharma et al. Dynamics-aware Unsupervised Discovery of Skills. ICLR 2020
>
> [b] Nachum et al. Data-efficient hierarchical reinforcement learning. NeurIPS 2018
>
> [c] Gupta et al. Relay Policy Learning: Solving Long-Horizon Tasks via Imitation and Reinforcement Learning. CoRL 2019
>
> [d] Pertsch et al. Accelerating Reinforcement Learning with Learned Skill Priors. CoRL 2020
>
> [e] Xie et al. Latent Skill Planning for Exploration and Transfer. ICLR 2020
>
> [f] Lu et al. Reset-Free Lifelong Learning with Skill-Space Planning. ICLR 2021
>
> [g] Wu et al. Example-Driven Model-Based Reinforcement Learning for Solving Long-Horizon Visuomotor Tasks. CoRL 2021
>
> [h] Shah et al. Value Function Spaces: Skill-Centric State Abstractions for Long-Horizon Reasoning. ICLR 2022

---

> > ### Comment · Reviewer_ctWV · 2022-08-25
> > **Thanks for the responses**
> >
> > I thank the authors for desponding in details to all the reviewers' comments.
> >
> > 1. Thank you for clarifying distinctions with prior skill learning work in greater detail. This section of the paper is much better now.
> >
> > 2. Regarding enabling long-term predictions, the authors mention that the main distinction from prior works like [e] is the learning of a skill-level dynamics model in addition to CEM planning at the level of skills. However, this is not quite convincing because there cannot be ground-truth supervision for the skill-dynamics model - there is no guarantee that the forward skill predictions always ends up in some state that is consistently meaningful. Specifically, I am still unclear about how having a skill dynamics model makes long term predictions *more reliable* compared to papers that plan at the level of skills without a skill dynamics model.
> >
> > 3. I am unfortunately not convinced by the points mentioned regarding comparison to hierarchical RL methods. This was also brought up by other reviewers. I understand that HIRO doesn't obtain high rewards based on the cited reference [g], but if the authors ran the DADS baseline, it would be helpful to incorporate the plots in the paper. Additionally, I still believe it is important to compare with at least one or two recent skill learning baselines that are very relevant (among those cited in [a] to [h] some of these for example [e], [g], and [h] are very closely related to the proposed algorithm and hence some comparisons would be helpful - it could even be cast as an ablation of the proposed algorithm and needn't be the exact algorithms in the respective papers)
> >
> > In light of the above concerns, and after reading the other reviews (reviewers also brought up the lack of real-robot experiments), I unfortunately still wouldn't lean towards acceptance.

---

> > > ### Author Response · Authors · 2022-08-27
> > > **Response to Reviewer ctWV’s Response**
> > >
> > > Thank you for your quick response and suggestions. In the revised paper, we have added DADS [a] and LSP [e] to Figure 4b and Section 4.2.
> > >
> > > &nbsp;
> > >
> > >
> > > **There cannot be ground-truth supervision for the skill-dynamics model.**
> > >
> > > The training of both a single-step model and skill dynamics model is **self-supervised learning**: for the single-step model, it takes $(s_t, a_t)$ as input and outputs $s_{t+1}$; whereas for the skill dynamics model, the input is $(s_t, a_t, a_{t+1}, \dots, a_{t+H-1})$ and the output is $s_{t+H}$, where the action sequence $(a_t, a_{t+1}, \dots, a_{t+H-1})$ is mapped into a skill latent $z_t$ using a skill encoder.
> > >
> > > We are not aiming for learning a task-specific skill dynamics model; so it is not required to predict any particularly important future state for a specific task. In other words, for the skill dynamics model, no state is inherently more meaningful than others. Thus, using the H-step future state as self-supervision is not only sufficient, but also ensures that the skill dynamics model is not biased for any particular downstream setting.
> > >
> > >
> > > &nbsp;
> > >
> > >
> > > **How having a skill dynamics model makes more reliable long-term predictions than a single-step model?**
> > >
> > > There are two main reasons: (1) the skill dynamics model enables accurate long-term prediction by significantly reducing the error accumulation, and (2) accurate future prediction improves CEM planning.
> > >
> > > 1. **Accurate long-term prediction**:
> > >
> > > Conventional single-step dynamics models have low long-horizon prediction accuracy because the model makes a small prediction error at every step, and this error compounds over time. A greater number of dynamics and policy predictions make long-horizon future predictions more error-prone.
> > >
> > > Our insight is that for long-horizon robotic tasks, it is more important to optimize the accuracy of long-horizon prediction than single-step details. Thus, we propose a skill dynamics model that “leaps ahead” to predict the resultant state of skill execution, instead of every intermediate state, which significantly reduces the number of necessary planning steps and error accumulation.
> > >
> > > For example, to predict the state 500 steps ahead, our approach requires 50 predictions of the skill dynamics model and high-level policy (with the skill length of 10), whereas a conventional single-step model requires 500 predictions of both the model and the low-level policy. We have empirically shown this quickly deviating from the ground truth, whereas the skill dynamics model makes reliable trajectory predictions in Figure 10 in Appendix.
> > >
> > > In summary, by reducing error accumulation, our skill dynamics model can predict long-term future states more accurately, which is important for complex robotic tasks.
> > >
> > > 2. **Better long-term planning**:
> > >
> > > The quality of CEM planning depends on the accuracy of the plan and its evaluation. The prior work LSP [e] plans in the skill space but the future state prediction is done by the single-step model, which is inaccurate as we discussed above, resulting in more erroneous evaluation of the plan. On the other hand, the skill dynamics model can provide more accurate future predictions, which leads to accurate evaluation of the plans and this results in better planning.
> > >
> > > &nbsp;
> > >
> > >
> > > **Incorporate the plots of DADS in the paper.**
> > >
> > > Thanks for your additional feedback. We have included the plot of DADS in FrankaKitchen in new Figure 4b. However, it is flat 0 and did not get any reward due to the challenges in acquiring meaningful skills in an unsupervised manner. We will include results on all other tasks in the camera-ready version.
> > >
> > > &nbsp;
> > >
> > >
> > > **Compare with recent skill learning baselines.**
> > >
> > > This is a great suggestion. Especially, LSP [e] is learning and planning on skills, but still using a single-step model to predict a future state. Thus, it is a good baseline to show why predicting the future with our skill dynamics model is beneficial.
> > >
> > > We have run the LSP code from the authors on FrankaKitchen. LSP succeeds in the first subtask very occasionally, and overall it does not learn meaningful behaviors (reward remains 0), whereas our method accomplishes 3-4 subtasks. We have included this result in Figure 4b and we will include an ablated method, SkiMo with a single-step model, to further investigate the benefit of our skill dynamics model.
> > >
> > > &nbsp;
> > >
> > >
> > > **Applicability to real robots**
> > >
> > > We do not have real robot experiments at present, but we believe we have made progress towards applying model-based RL to more complex real-world robotic tasks via improving sample efficiency. Our extensive experiments suggest a high potential for SkiMo in real-world applicability. Please refer to our response to the meta-reviewer for detailed explanations.
> > >
> > > &nbsp;
> > >
> > > Thanks again for your valuable feedback, which has helped us improve our paper. Please let us know if this addresses all your concerns and if there is any further concern that potentially prevents you from accepting this paper.

---

> ### Author Response · Authors · 2022-08-24
> **Follow up on response**
>
> Thank you for your valuable comments and feedback. We hope our rebuttal has addressed all your concerns. Please let us know if you need any further clarification! We would love to answer your questions and make our paper stronger!

---

### Official Review · Reviewer_6Ygv · 2022-07-29

**Originality:** Very Good
**Technical Quality:** Very Good
**Clarity Of Presentation:** Excellent
**Impact:** 4

**Recommendation:**

Weak Accept: I recommend accepting the paper, but will not argue for my recommendation if the majority of other reviewers have a different opinion.

**Summary:**

This paper presents an algorithm for skill based model based RL, where the model is not learned as a mapping from state and action to next state but over outcomes of performing skills. The model is learned with a VEA style loss where a skill embedding is learned and a dynamics model in this embedding space is learned together with the skill from offline data. This setup is then used to learn a downstream skill for a new task using the learned dynamics model in a MBRL way. The paper presents experiments on a maze task and a franka kitchen environment where the presented method outperforms the baseline methods. The paper also presents ablation studies.

**Issues:**

see above under weaknesses.

**Quality Of The Limitations Section:**

Additional details required

**Reviewer Expertise:**

4: The reviewer is confident but not absolutely certain that the evaluation is correct

**Robotics Focus:**

Highly relevant to robotics but no hardware experiments

**Strengths And Weaknesses:**

Strengths:

I think the paper is well written and presents clearly the problem setting, the algorithm and the contribution. I also think the authors are making clear how their approach differs from the related work. The topic and the direction of the paper is very relevant of learning models over higher abstractions instead of low-level sensor measurements and the paper presents convincing results that this might be a good avenue to investigate further. I think this paper would make a clear contribution to the field.

Weaknesses:
- it would be very good and important to know more about the offline data that was used, how close is it to the downstream tasks? How much data was used? I think these details are important to understand the generality of this approach

- robot experiments are missing, how would the transfer look like? still your method requires at least 250k samples, even after all of the pre-training that happened - could you maybe elaborate on this and comment on the real world applicability of the approach?
minor:
- an overview image of the algorithm would be very helpful to understand quicker how the learning setup looks like.

**Summary Of Recommendation:**

I am willing to adjust my assessment during the rebuttal after reading the authors response on the above mentioned points.

---

> ### Author Response · Authors · 2022-08-19
> **Response to Reviewer 6Ygv**
>
> Thank you for your constructive feedback. We address your concerns in detail below and have updated our paper accordingly.
>
> &nbsp;
>
>
> **More information about the offline data: how close is it to the downstream task? How much data was used?**
>
> As per Reviewer 6Ygv’s suggestion, we have added more details about the offline data in Appendix, Section C.3. In short, we use task-agnostic data (600-3000 trajectories per environment) whose distribution can be different from the downstream tasks.
>
> For instance, in the Kitchen dataset, demonstrations of each subtask are present in the data, but the agent needs to recombine these skills in a new way to solve the task. The CALVIN dataset is especially task-agnostic: each subtask transition has probability lower than 0.1% on average in data, challenging the agent to extract general skills and adapt to the downstream task distribution.
>
> &nbsp;
>
>
> **Applicability to real robots**
>
> Our method aims to **improve sample efficiency**, which is a major bottleneck for applying learning approaches to real-world robots. We demonstrate that SkiMo consistently outperforms baselines in **four different tasks**, and this suggests a high potential for SkiMo in real-world applicability. Applying SkiMo to real robots is our immediate future work!
>
> We would like to highlight that a large number of online interactions (e.g. 250k) used in our experiments is mainly due to the high complexity of the tasks we considered. We believe our method requires much fewer samples (even a few trials) to learn simpler tasks. Further, even in such complex long-horizon tasks, our method achieves 5x fewer samples than baseline approaches, which implies its advantage in real-world robot learning.
>
> In terms of training stability and complexity, a large portion of our approach can be pre-trained offline, and both pre-training and downstream RL training phases have been tested in diverse environments and tasks, as explained in the previous response. Particularly, Dreamer (the base RL algorithm of our method) has recently proved its applicability to real-world robots in DayDreamer [a] by training a quadruped robot to walk only with one hour of real-world experience. Given that our method is a more sample-efficient version of Dreamer using skills, we are confident that our approach can be applied to real-world robots as well.
>
> In addition, our method can **guide safe exploration**, which is critical in real-world robot learning, as it does not explore random actions but try only meaningful skills extracted from offline data. We further discussed the applicability to real robot systems in Appendix, Section D.
>
> &nbsp;
>
>
> **An overview image of the algorithm would be very helpful.**
>
> Great suggestion! We have added Figure 11 in Appendix to illustrate our training procedure.
>
>
> &nbsp;
>
>
> **References**
>
> [a] Wu et al. DayDreamer: World Models for Physical Robot Learning. arXiv 2022

---

> > ### Comment · Reviewer_6Ygv · 2022-08-25
> > **Response to Author response**
> >
> > I would like to thank the authors for their response and for clarifying and addressing my concerns, after reading the answers also to the other reviewers I would like to increase my score.

---

### Author Response · Authors · 2022-08-19
**Revised PDF for rebuttal**

We thank all reviewers for their constructive feedback and for helping us make our paper a stronger submission!

First of all, we highlight that we have included two additional tasks, **CALVIN** and **FrankaKitchen (misaligned task)**, in the supplementary materials. In short, our method consistently shows **significant improvement in sample efficiency** over prior works on **4 different tasks**, reassuring us about our method’s robustness and generality.

We have revised our paper to address reviewers’ comments. The major changes listed below are highlighted in red:
* Section 4: Add two additional tasks: CALVIN and FrankaKitchen (misaligned task)
* Section 4: Add two new baselines: TD-MPC and SPiRL+TD-MPC
* Section 3: Re-organize and clarify the method section
* Section 2: Discuss differences between our paper and prior work on model-based RL with skills and hierarchical RL
* Appendix, Section A.3: Add an ablation study on fine-tuning the skill dynamics model
* Appendix, Section C.3: Add details about offline data
* Appendix, Figure 11: Add an overview figure of the algorithm

We temporarily attached the appendix to the main paper for your convenience.

We hope we have addressed all your concerns and questions. Please let us know if there are any concerns preventing you from raising your score.

---

### Comment · Area_Chair_anUx · 2022-08-25
**Last chance for reviewer responses to author replies**

Dear reviewers, the authors have responded in detail to the reviews.
Please do further engage with the authors now, if possible, as the author/reviewer discussion/rebuttal phase ends Aug 27 at 11:59 PM Pacific.

Many thanks for the two reviewers that have already responded to the author's responses,
and many thanks in advance to those still needing a bit more time to respond --
your participation greatly contributes to the overall quality and value of the review process!

-- your Area Chair

---

### Meta-Review · Area_Chair_anUx · 2022-08-15

**Recommendation:** Accept (Poster)
**Confidence:** 3

**Metareview:**

A method is proposed for skill based model based RL. Rather than learning a mapping of the
discrete-time dynamics, the outcomes of performing skills are learned. Learning of downstream tasks
is then demonstrated, along with ablation studies.

The four reviews currently have mixed opinions:  weak accept (x2), weak reject (x2).
In favor is the belief that the community can benefit from this work towards better learned temporal abstractions that can be leveraged for better long-term planning.  Against is that this paper revisits temporal abstraction ideas with a complex method (many learned pieces), and that there is (arguably) no core idea that would be of general use to the wider robotics community.
Overall, I lean in favor of letting time judge whether the method will be useful in the long run.
My recommendation:  Accept (Poster).

Strengths:
- the problem is relevant to many robot learning settings
- well written, presents the problems setting / algo / contributions very clearly.
  Differences with prior work, e.g., Dreamer, are well explained.
- performance gains seem impressive on the tasks presented;
  extensive appendix with additional experiments

Weaknesses:
- the distinction with prior MBRL work that learns a skill space for conditioning a policy, is not clear.
- important to compare with several external hierarchical RL approaches proposed in the literature over the years
  i.e., HIRO, DADS.
- very complex model: 10 objective function terms, 9 networks / model components; difficulty to reproduce;
  The relative importance / weighting of the various loss terms needs to be discussed
- no real robot experiments;  demonstrated on only two simulated tasks
- organization & definitions in Section 3.2 needs revisiting
- further details on the offline data are needed
- the policy could in principle still be ignoring the skill that it is conditioned on


**Best Paper Nomination:**

No

---

> ### Author Response · Authors · 2022-08-19
> **Response to Meta Reviewer (1/2)**
>
> We thank the area chair for a thorough summary. We first address the common concerns of the reviewers here.
>
> &nbsp;
>
>
> **Additional experiments on two new tasks: CALVIN and FrankaKitchen (different task ordering) [43gq]**
>
> We highlight that we have added two additional manipulation tasks to the supplementary materials (now merged into Section 4). These additional experiments demonstrate consistent results with the ones in the main paper, reassuring us about our method’s robustness and generality. In summary, our method consistently shows **significant improvement in sample efficiency** over prior works on **four** different tasks: Maze, FrankaKitchen, FrankaKitchen (Mis-aligned task), and CALVIN.
>
> We hope our experimental results on four different tasks provide sufficient evidence of the potential transferability of our method to the real world. Demonstrating our method in the real world is our definite future work!
>
> In addition, we have updated results with two more baselines ”TD-MPC” and ”SPiRL+TD-MPC” in the revised version, which ensures that the improvement of our method is not due to the choice of model-based RL algorithms.
>
> &nbsp;
>
>
> **Complexity of the proposed approach [43gq]**
>
> Although our method, SkiMo, looks complicated at a glance, its implementation leverages a simple combination of well-explored approaches, such as Dreamer, SPiRL, and CEM. In other words, each component of our method has been widely verified and thus, easy to get reproduced.
>
> Moreover, 4 out of 9 components (observation decoder, skill encoder, skill prior, skill policy) are **entirely pre-trained using supervised learning**, which is easy to train and is done offline. The online RL training phase matches the complexity of typical model-based RL (e.g. Dreamer and TD-MPC), which trains only 5 **essential** components (state encoder, dynamics model, reward function, value function, and task policy).
>
> Finally, we believe that the number of components and loss terms are not metrics to measure whether the method is useful or whether it is reproducible. We also provided the code and data as supplementary materials for reproducing the results.
>
> &nbsp;
>
>
> **Applicability to real robots [6Ygv, 43gq]**
>
> Our method aims to **improve sample efficiency**, which is a major bottleneck for applying learning approaches to real-world robots. We demonstrate that SkiMo consistently outperforms baselines in **four different tasks**, and this suggests a high potential for SkiMo in real-world applicability. Applying SkiMo to real robots is our immediate future work!
>
> In terms of training stability and complexity, a large portion of our approach can be pre-trained offline, and both pre-training and downstream RL training phases have been tested in diverse environments and tasks, as explained in the previous response. Particularly, Dreamer (the base RL algorithm of our method) has recently proved its applicability to real-world robots in DayDreamer [a] by training a quadruped robot to walk only with one hour of real-world experience. Given that our method is a more sample-efficient version of Dreamer using skills, we are confident that our approach can be applied to real-world robots as well.
>
> In addition, our method can **guide safe exploration**, which is critical in real-world robot learning, as it does not explore random actions but try only meaningful skills extracted from offline data. We further discussed the applicability to real robot systems in Appendix, Section D.
>
> &nbsp;
>
>
> **Weighting of different loss terms [ctWV, 43gq]**
>
> We thank the reviewers to point out the missing weighting factors of loss terms. To make this point clear, we explicitly wrote the weighting factors in Equation 2-6 in the revised version.
>
> We simply take the weighting factors from prior work (SPiRL, Dreamer, and TD-MPC) and they work well across four different navigation/manipulation tasks. The full list of hyperparameters can be found in Appendix, Table 2.
>
> &nbsp;
>
>
> **Organization and clarification of the method section [43gq, gSBE]**
>
> We appreciate the suggestions and have revised the method section accordingly.
> * We further explained each variable and model component in Section 3.1 and 3.2.
> * To avoid confusion, we explicitly specified that $\psi, \theta, \phi$ are sets of model parameters in Section 3.2.
> * We revised Section 3.3 and 3.4 for clarity and corrected typos and errors.

---

> > ### Author Response · Authors · 2022-08-19
> > **Response to Meta Reviewer (2/2)**
> >
> > **Distinction with prior model-based RL work that uses skills [ctWV]**
> >
> > We have added more discussion about prior model-based RL work with skills to related work in the revised version.
> > * Prior model-based RL approaches [b-d] also plan over skills. However, their dynamics models make **single-step predictions** (conditioned on the skill latent) as conventional model-based RL, which struggles at handling long-horizon planning due to error accumulation.
> > * Wu et al. [e] proposes to learn a temporally-extended dynamics model; however, it conditions on low-level actions rather than skills and is only used for low-level planning.
> > * A concurrent work, Shah et al. [f], is most similar to our paper in that it learns a skill dynamics model, but it uses a limited set of discrete, manually-defined skills.
> >
> > Instead, we devise a skill-level dynamics model that learns to make accurate long-term predictions with a jointly extracted skill library for scalable long-horizon learning.
> >
> > &nbsp;
> >
> >
> > **Comparison to hierarchical RL approaches [ctWV]**
> >
> > We thank the reviewers for suggesting the highly relevant works, DADS [b] and HIRO [g], which propose to efficiently learn complex tasks using hierarchical policies. We have added these papers to related work.
> >
> > We conducted additional experiments and found that **DADS could not get any reward on FrankaKitchen**. DADS learns diverse behaviors but they do not contain meaningful skills. Similarly, Figure 5a in Gupta et al. [h] verifies that **HIRO fails to learn FrankaKitchen** (i.e. gets near 0 reward).
> >
> > We would like to emphasize that we instead compared our method against a strictly stronger hierarchical RL method, SPiRL [i], which leverages large offline data, whereas DADS and HIRO learn hierarchical policies solely online.
> >
> > &nbsp;
> >
> >
> > **The policy may ignore the skill latent [ctWV]**
> >
> > We assume the offline task-agnostic data contains diverse behaviors. This multi-modality of data enforces a low-level skill policy not to ignore the skill latent $z$ to diversify its behaviors given a state. Otherwise, the policy cannot reconstruct the correct behavior among diverse trajectories given a state in the dataset, which results in a **high reconstruction loss** in VAE training.
> >
> > More technically, we train the skill policy using $\beta$-VAE, where $\beta$ controls how much the skill latent affects the policy. Thus, we can ensure that the policy utilizes the skill latent with a small enough $\beta$.
> >
> > &nbsp;
> >
> >
> > **Details about the offline data: how close is it to the downstream task? How much data was used? [6Ygv]**
> >
> > As per Reviewer 6Ygv’s suggestion, we added more details about the offline data in Appendix, Section C.3. In short, we use task-agnostic data (600-3000 trajectories per environment) whose distribution can be different from the downstream tasks.
> >
> > For instance, in the Kitchen dataset, demonstrations of each subtask are present in the data, but the agent needs to recombine these skills in a new way to solve the task. The CALVIN dataset is especially task-agnostic: each subtask transition has probability lower than 0.1% on average in data, challenging the agent to extract general skills and adapt to the downstream task distribution.
> >
> > &nbsp;
> >
> >
> > **What is “sg” in Equation 4 and 7? [43gq, gSBE]**
> >
> > Thank you for spotting this. “sg” denotes the “stop gradients” operator, which does not back-propagate gradients to the skill encoder $q_\theta$ in Equation 4 and the state encoder $h_t=E_\phi(s_t)$ in Equation 7. We added the definition of “sg” after Equation 4.
> >
> > &nbsp;
> >
> >
> > **References**
> >
> > [a] Wu et al. DayDreamer: World Models for Physical Robot Learning. arXiv 2022
> >
> > [b] Sharma et al. Dynamics-aware Unsupervised Discovery of Skills. ICLR 2020
> >
> > [c] Xie et al. Latent Skill Planning for Exploration and Transfer. ICLR 2020
> >
> > [d] Lu et al. Reset-Free Lifelong Learning with Skill-Space Planning. ICLR 2021
> >
> > [e] Wu et al. Example-Driven Model-Based Reinforcement Learning for Solving Long-Horizon Visuomotor Tasks. CoRL 2021
> >
> > [f] Shah et al. Value Function Spaces: Skill-Centric State Abstractions for Long-Horizon Reasoning. ICLR 2022
> >
> > [g] Nachum et al. Data-efficient Hierarchical Reinforcement Learning. NeurIPS 2018
> >
> > [h] Gupta et al. Relay Policy Learning: Solving Long-Horizon Tasks via Imitation and Reinforcement Learning. CoRL 2019
> >
> > [i] Pertsch et al. Accelerating Reinforcement Learning with Learned Skill Priors. CoRL 2020